# Investigation of Lexical and Inflectional Verb Production and Comprehension in French-Speaking Teenagers with Developmental Language Disorders (DLDs)

**DOI:** 10.3390/bs15091252

**Published:** 2025-09-14

**Authors:** Marie Pourquié, Emilie Courteau, Ann-Sophie Duquette, Phaedra Royle

**Affiliations:** 1Department of Linguistics and Basque Studies, University of the Basque Country (UPV/EHU), 01006 Vitoria-Gasteiz, Spain; 2Department of Psychology & Neuroscience, Dalhousie University, Halifax, NS B3H 4R2, Canada; emilie.courteau@dal.ca; 3School of Speech-Language Pathology and Audiology, Faculty of Medicine, University of Montreal, Montreal, QC H3C 3J7, Canada; asduquette@foniaorthophonie.com (A.-S.D.); phaedra.royle@umontreal.ca (P.R.); 4Centre for Research on Brain, Language and Music, Montreal, QC H3G 2A8, Canada

**Keywords:** DLD, teenagers, verb agreement, production, comprehension, French, language assessment clinical tools

## Abstract

Little research has studied verb inflection and argument structure complexity effects in teenagers with developmental language disorders (DLDs). However, verb production and comprehension deficits that characterize younger children with DLD might persist over time. Seventeen French-speaking teenagers with DLD and seventeen controls (typical language, TL group) were tested with fLEX, an application designed to assess lexical and inflectional production and comprehension of three different verb types: intransitives, transitives and ditransitives, i.e., verbs that require none, one or two overt complements. Participants performed three tasks: action naming, sentence production and sentence comprehension involving third singular and plural present tense. Both groups performed similarly on action naming. Subject–verb agreement errors characterized participants with DLD both in sentence production and comprehension; however, verb–argument structure had no effect on any of the tasks. These results characterize verb deficits in teenagers with DLD as affecting inflectional processes rather than lexical ones: they are found in production and comprehension, persist until adolescence and are thus a target for evaluation and intervention in French-speaking teenagers. Results are discussed from a cross-linguistic perspective and in light of current theories on DLD.

## 1. Introduction

In accordance with recent guidelines, we use the term developmental language disorders (DLDs) where “disorders” is used to refer to conditions without obvious etiology ([23]) and to refer to what was previously known as Specific Language Impairment (SLI). Developmental language disorders (DLDs, [8]) are often described as developmental impairment linked to linguistic disabilities. However, it is not the case that the whole of language competence in this population is equally impaired. Children with DLD often show relative strengths in lexical–semantics but difficulties in morphosyntax or phonology ([67]). Furthermore, while verb inflection deficits have been reported in many languages and considered as a characteristic feature of DLD, it is not clear whether they persist in both production and comprehension in teenagers ([16]). Also, little research has studied verb inflection and argument structure complexity effects in DLD ([29]) and especially in teenagers. In the following sections, cross-linguistic manifestations of DLD are described specifically regarding verb production and comprehension deficits in preschool-aged children and teenagers. First, we review studies targeting DLD in preschool-aged children and then we turn to the few studies on grade-school-aged children and teenagers with DLD.

### 1.1. Verb Inflection Production and Comprehension Deficits in Preschool-Aged Children with DLD

A majority of studies on verb inflection in children with DLD involve verb production in English. They have mainly focused on the past tense (see [7] for a review of English studies). However, other languages have been investigated, including non-Indo-European languages (see, e.g., [39]) and Romance languages such as Italian and French (see, e.g., [62]). Most studies on subject–verb (SV) number agreement focus on production, using spontaneous speech, or elicitation methods in various languages ([9]).

A seminal study of English-speaking children with DLD aged 3;8–5;7, including age-matched and language-matched controls (aged 2;11–3;4), evaluated spontaneous speech errors and probed elicitation of inflection including SV agreement on singular verbs and copulas ([40] ([40])). Children with DLD performed worse on these items than both control groups. In the same study, Italian-speaking children with DLD aged 4–5;11, including age-matched and language-matched controls (aged 2;6–3;6), exhibited differences in third person *plural* agreement but not third person *singular* verbs. [1] ([1]) found that spontaneous speech production in speakers of Urban Hijazi Arabic (Saudi Arabia) with DLD aged 4:0–5;3 contains more third person feminine singular and plural SV, but not masculine agreement errors, than their age-matched and language-matched peers (2;5 years old on average); however, these errors are less common than gender and person ones. Two studies in Finnish and Hebrew did not reveal significant difficulties on SV agreement in children with DLD. In elicited inflected-verb production using probes (images and videos) with Finnish-speaking children aged 5;2, [34] ([34]) found that children with DLD do numerically worse (93.7%) than both age-matched and language-matched controls (aged 3;8 on average) who scored high (99.7%). However, tense-marking errors were more salient than number-agreement ones. Using sentence-completion probes, [20] ([20]) found that Hebrew-speaking children with DLD aged 4;1–5;11 had more difficulty than age-matched and language-matched controls (aged 2;9–4;0) producing appropriately agreeing present and past tense verbs. However, errors were linked to unstressed markers for verb inflection patterns rather than singular–plural distinctions.

Comparing two languages within the same child, [65] ([65]) presented spontaneous speech samples and probes for SV agreement from a Greek–French 9-year-old bilingual child and two bilingual French-dominant controls matched on Greek and French language age, aged 4;7 and 5;11 respectively. The participant with DLD produced more verb agreement errors in French than in Greek and was only significantly different with the older control in Greek. However, studies of Greek preschool children with DLD disagree on whether SV agreement is impaired, and agreement mastery appears to be linked to task and age (see [35]).

In slightly older French-speaking children, [49] ([49]) found no differences in SV agreement between children with DLD aged 7;6 compared to age-matched and language-matched controls aged 3;3. Only tense-marking revealed differences with age-matched groups. In a follow-up study on Italian, [10] ([10]) found that SV agreement, together with determiner production, provides a sensitive measure for the identification of Italian-speaking children with DLD aged 4;1–7. Overall, cross-linguistic studies on DLD manifestations report variable verb inflection deficits related to SV agreement and tense.

A small number of studies have focused on SV agreement comprehension. Importantly, this aspect of language is central to our understanding of language abilities in children with DLD. In the study by [40] ([40]) presented above, Italian children with DLD (aged 3;8–5;7) were also probed on sentence comprehension using a forced-choice sentence–picture matching task with four probes (including a target, a number error foil, and two lexical foils). The group with DLD, although above chance, made more errors than both control groups on plural *and singular* verbs in the third person, in contrast to production data where singulars appeared to be mastered.

[20]’s ([20]) study on Hebrew-speaking five-year-olds presented above also assessed verb comprehension (focusing on gender or number) using a forced-choice sentence–picture matching task with three probes (the target, a gender error foil, and a number error foil). They found that children with DLD (aged 4:1–5;11) had more difficulty than age-matched peers on a present tense third person comprehension task. No details are given about which errors were observed.

### 1.2. Verb Inflection Deficits in Grade-School-Aged Children and Teenagers with DLD

Some studies have reported verb agreement deficits in grade-school-aged children with DLD speaking various languages. In English, [28] ([28]) observed a preference for unmarked singular forms in a case study of a French–English bilingual boy aged 8–9 years in both spontaneous speech (e.g., *The ambulance *arrive*) and sentence repetition. In French, [27] ([27]) find that children with DLD aged 5;4 to 9;4 have difficulty completing sentences involving singular and plural third person SV agreement of the verb *faire* ‘to do’ (an irregular verb): the singular remains the default in participants with DLD, even though *faire* is arguably one of the most frequent verbs in their language and thus subject to repeated input and practice. In German-speaking children with DLD aged 5:8–7:11 and English-speaking ones aged 10–13;1, [14] ([14]) observe low levels of correct third person singular -*s* marking in English participants (below 50% on average). These results contrasted with past tense marking, which was relatively better (above 75%). Similarly, in German, SV agreement was low at 64% correct on average, while *sein* ‘to be’, a suppletive (irregular) verb analyzed separately, was at 88% correct. A study of Hungarian by [42] ([42]) compared verb production in children with DLD aged 9;10 and vocabulary-matched controls (aged 7;1). They observe significant group effects which were found mainly on second person plural forms. [30] ([30]) find more SV agreement errors in the spontaneous speech of Turkish-speaking children with DLD aged 4;0 to 7;10 (*M* = 5;3) in comparison to age-matched and MLU-matched peers (aged 2;10 on average). Children were more likely to produce an SV error on the verb if the morpheme was irregular; however, word length in phonemes was a better predictor for errors. [31] ([31], in [22]) showed that Dutch-speaking children with DLD older than 8 years reveal persistent problems in spontaneous language samples, particularly errors with SV agreement. Coherent with [27] ([27]), who used a single verb to probe number agreement, [59] ([59]) showed difficulties producing inflected verbs in an elicitation task. They elicited past tense and singular and plural SV agreement in French participants with DLD aged 9–46. We know of no other study of French verb number–agreement comprehension abilities in (pre-)teens.

A few studies have focused on verb production and comprehension deficits in teenagers, i.e., in children older than 10 years old. Interestingly, another study with Dutch teenagers with DLD (mean age 12;7) by [70] ([70]) reveals some improvement but persistent subtle deficits in verb inflection production: children with DLD performed better on SV agreement tasks than a younger DLD group (aged 6;1–8;0). Their performance was around 95%, while typically developing children reach a ceiling around 6 years of age. [70] ([70]) highlight that errors were dependent on the linguistic context: more errors were made in main clauses than embedded ones, possibly linked to verb second movement.

[45] ([45]) showed that English teenagers aged 16 with DLD are less sensitive to the omission or substitution of grammatical morphemes than teenagers with TD language. They exhibit verb inflection production and comprehension deficits in both third singular agreement and past tense. [35] ([35]) found that Greek-speaking children with DLD aged 8;5 on average do not show significant difficulties producing SV agreement during an elicitation task, but differ from controls (aged 9;7 on average) in a grammaticality judgement task where children with DLD were on average at 56% (SD = 32) versus 99% in controls (SD = 2). These results highlight the importance of contrasting comprehension and production to better understand linguistic abilities in children with DLD. [22] ([22]) argue that both receptive and expressive tasks should be used to test whether there is a gap between knowledge (i.e., comprehension) and performance (i.e., production) in DLD.

The reviewed studies have reported the presence of verb inflection deficits in grade-school-aged children and teenagers with DLD, suggesting that these deficits can persist in DLD; however, some data show no salient difficulties. More studies and in diverse languages are necessary to determine whether they persist in both the production and comprehension of verb inflection.

### 1.3. Effects of Verb Argument Structure Properties in DLD

Despite the abundant literature on verb deficits in DLD across languages, little attention has been paid to argument structure deficits in this population ([19]). In fact, it has been claimed that thematic relations or argument structure are intact (e.g., [55]). However, it has been reported that argument structure repertoires are less diverse and complex in young children with DLD than in typically developing (TD) children ([25]). Furthermore, it could be that observed morphological errors are linked at least partly to argument structure complexity ([29]; [51]; [64]). Reported errors related to verb argument structure properties typically concern the omission of obligatory complements (e.g., direct objects), the omission of auxiliary verbs and a higher number of morphological errors correlated with argument structure.

[51] ([51]) teased apart lexical–semantic knowledge and morphosyntactic production by comparing action naming and sentence production. They compared the production of French sentences of the same length but with different argument structures (e.g., transitives vs. intransitives with adjuncts such as *Le canard écrit une lettre* ‘The duck writes a letter’ vs. *Le canard court dans les champs* ‘The duck runs in the fields’, respectively). In verb naming and in verb–picture matching tasks, no effect of argument structure was found for children with DLD or TD (mean age 9;11 and 5;11, respectively; the TD group was matched on sentence production abilities). In contrast, during sentence production, children with DLD made more errors (especially auxiliary omissions) than TD children in transitive than in intransitive sentences. Because sentences were all of the same length, errors were interpreted to be linked to verb–argument structure.

[64] ([64]) investigated SV agreement and argument structure production in spontaneous speech, comparing monolingual Dutch-speaking children with DLD and bilingual Frisian–Dutch children with DLD (ages 4;07–7;04; mean 5;11) to monolingual Dutch TD children (ages 2;07–3;06; mean 3;02). Both SV agreement and argument structure deficits characterized DLD in monolingual and bilingual Dutch-speaking children. Furthermore, a link between omission errors and verb complexity was observed in children with DLD but not in TD children, with transitive sentences being more difficult than intransitive ones. The authors concluded that SV agreement and argument structure were both affected in children with DLD. However, in the study presented above, [30] ([30]) found that verb transitivity did not impact agreement errors in Turkish-speaking children with DLD aged 4;0 to 7;10 (*M* = 5;3).

Taking into account verb–argument structure in DLD assessment has been claimed to be necessary from a clinical perspective ([24]; [51]; [19]; [3]). Language processing deficits in DLD related to verb argument structure complexity have been reported across languages (e.g., English: [29]; [24]; French: [51]; Dutch and Frisian: [64]; [19]; Spanish: [2]); however, no consensus has been found regarding whether argument structure deficits result from lexical or morphosyntactic impairment.

### 1.4. Accounts of Verb Production and Comprehension Deficits in DLD

Several accounts pointing toward an agreement processing impairment in DLD assume that both production and comprehension are impaired. The Missing Agreement Account proposed by [13] ([13]) suggested that DLD children have problems establishing structural relations between two elements where one asymmetrically controls the other, as in SV agreement. This was challenged by [56] ([56]) who noted that German-speaking DLD children correctly produced finite verbs on which the verb movement was successfully applied, as long as SV agreement was mastered. They argued that agreement deficits were better explained by the Extended Optional Infinitive account ([57]), which proposed that finiteness markers are omitted for an extended period in children with DLD compared to TD children. [48] ([48]) proposed the Extended Optional Default Account to explain the fact that, in utterances by French-speaking children with DLD, there was an overuse of the present tense finite verb stem that appeared to have the status of a root infinitive. [46] ([46]) proposed the Fragile Computation of Agreement hypothesis. They argue that agreement difficulties in children with DLD are related to the structural distance that separates elements involved in agreement relations and that their deficits increase as a function of the complexity of this agreement configuration. These authors underline the advantage of using a comprehension task and assume that the SV agreement difficulties characterizing children with DLD “may be linked to a specific impairment in the abstract grammatical computation, which underlies both production and comprehension” (p. 934).

[21] ([21]) noted that agreement production in Dutch-speaking teenagers with DLD does not improve when compared to a younger population, but their judgment on agreement becomes more accurate, which shows that knowledge can develop in the presence of impaired performance. This author pointed out that [6]’s ([6]) Vulnerable Marker hypothesis takes into account this dichotomy between comprehension and production: it suggests that processing limitations prevent children with DLD from mastering agreement production in the presence of developed knowledge. Other resource-based accounts, such as the Surface Hypothesis ([38]), propose that acoustic properties of agreement markers will influence their mastery, or that children with DLD will focus their limited resources on salient grammatical cues (the Morphological Richness Account, [11]). However, cross-linguistically, these hypotheses are not directly supported due to differences in the acoustics or salience of grammatical cues. For example, in English, as in Italian, marked forms seem to be the most difficult to produce despite important differences in how they are realized across these two languages ([40]). Regarding French, our language of interest, it is important to note that inflection is always syllabic (contrary to English) and on the most stressed syllable on the verb (also contrary to English). However, as we shall see, number marking on the verb can be quite irregular in the oral form.

Van der Lely’s Representational Deficit for Dependency Relations hypothesis (RDDR) proposes that the cause of DLD lies in the computational syntactic system ([69]). The syntactic module is underspecified with respect to the relationships between constituents. Children with SLI lack a specific syntactic principle required to process agreement. The deficit is at the level of the linguistic representation, in the computational system, and this is the reason why this hypothesis is considered a representational account. [32]’s ([32]) Computational Complexity Hypothesis does not locate the deficit at the computational system itself but rather at the interfaces. Producing grammatical outputs is taxing and gradual depending on the number of computations involved in the production or comprehension of language ([43]).

[68] ([68]) observed specific difficulties in grammar (including phonology) in individuals with DLD, in the presence of relative lexico-semantic strengths, thus prompting the Procedural Deficit Hypothesis where aspects of grammar learned through procedural memory, and local dependencies, such as argument structure, would remain relatively intact as they are believed to be aspects of language that rely on declarative memory.

### 1.5. The Present Study

To our knowledge, the effects of verb argument structure properties on action naming, sentence production and comprehension in teenagers with DLD have not been reported nor investigated. Also, while verb inflection deficits have been reported in many languages and considered as a characteristic feature of DLD, it is not clear whether they persist in both production and comprehension in French-speaking teenagers. It is also not clear whether the persistent language deficits that characterize teenagers with DLD show the same relative strengths in lexical–semantics and difficulties in morphosyntax as reported in younger children with DLD. The goal of the present study is to assess whether both SV agreement production and comprehension deficits are found in French teenagers with DLD and whether difficulties increase with verb argument structure complexity. Since SV number agreement errors are produced by TD children during acquisition ([27]) but are no longer expected to be produced by French-speaking teenagers, assessing teenagers enables us to clearly disentangle atypical from typical language processing in teenagers with DLD. If persons with DLD do in fact have persistent lexical deficits, teenagers with DLD should exhibit more difficulties than neurotypical teenagers in verb naming, sentence production and sentence comprehension tasks. On the other hand, if the source of their difficulties is based on inflectional processes both in the expressive and receptive domains, no difference is expected between both groups on the verb naming task which does not require verb inflection; however, verb agreement errors are expected to be made in sentence production and comprehension tasks. Furthermore, in comprehension tasks, teenagers with DLD should specifically make more morphosyntactic errors than controls.

## 2. Materials and Methods

### 2.1. Participants

A total of 34 pre-teens and teenagers participated in this study. They were French-speaking (French monolingual or first and main language at home and in school) and taking part in a larger study on neurocognitive functions in DLD (see [16], for data on other cognitive and linguistic tasks and [17] for ERP data). The protocol was approved by the University of Montreal Research Ethics Board for educational and psychology research (CERES). All participants’ parents gave written consent for their child’s participation prior to the experimental session. All had a hearing screening on the first day of assessment (500 Hz to 8000 Hz at 25 dB in at least one ear). Their first language was French, as assessed by a demographic questionnaire, and it was also their language of instruction and daily use (minimum = 95%). All had normal or corrected-to-normal vision and had no major illnesses or a history of prolonged hospitalization (myringotomy, tympanic tube placement and tonsillectomy were not exclusionary criteria).

#### 2.1.1. Participants with DLD

Seventeen participants with DLD (10 girls), aged between 12 and 15 years (*M* = 14;06; *SD* = 0.68), were recruited from a specialized school for French children and teenagers with learning disabilities in Montreal (Québec, Canada). This school excludes children with disruptive behavior which possibly explains why our group with DLD includes more girls than boys. The remaining participants with DLD were recruited from a parents’ association for children with DLD. All participants had a documented history of DLD, with a complete speech language pathologist’s language evaluation (including narrative and pragmatic domains) resulting in a diagnosis. Note that many participants had co-occurring disorders, such as ADHD and dyspraxia. These disorders are often co-morbid with DLD and do not preclude a DLD diagnosis (see *Statement 9*; [8]). The dominant clinical profile was the presence of persistent language difficulties. All participants had been diagnosed before kindergarten or during the first year of primary school, and maintained significant functional impairments needing adaptations to succeed in school. Additionally, the DLD group had significantly lower scores on the recalling sentence task (CELF-IV^cnd-F^, French version, [63]) than the TD group (see Table 1). Recalling sentences has been shown to discriminate between typical and disordered language development in many languages including French in children and adults ([36]; [52]).

#### 2.1.2. Control Participants

Seventeen participants with no history of language impairment, i.e., typical language (TL group) were taken from a pool of twenty TL participants and were matched with DLD participants on gender and language background (French monolingual or first and main language at home and in school). They were aged 10–14 (*M* = 12.76, *SD* = 1.49). Their typical language development was established via a parental questionnaire filled out during a phone interview, as well as confirmed by our language and cognitive tasks (see Table 1). Language abilities were assessed through the recalling sentence and word class receptive tasks (CELF-IV cnd-F, French version, [63]). Initially, an attempt was made to compare data to a second control group matched on language abilities. Based on their performance in the sentence repetition task, this would have obliged us to recruit 7-year-old participants for more than 30% of the DLD participants. Knowing that participants should be at least 8.5 years old to properly perform SV agreement, we decided to use only similar aged controls in our TL group. The two groups were otherwise matched on nonverbal skills using tasks from the Cognitive Experiments IV v2 package of Presentation^®^ software (version 18.0, Neurobehavioral Systems, Inc., Berkeley, CA, www.neurobs.com). Nonverbal working memory was measured using the Corsi forward and backward block tasks ([15]) and a delayed match-to-sample of nonverbal stimuli ([18]) with 1 or 5 s delays. To compare groups, we used Brunner–Munzel tests ([12]), as recommended by [58] ([58]) for skewed data with small sample sizes. Despite matching, differences between groups were found in age (1.5 years on average) and schooling (1 year on average), to the advantage of participants with DLD, on the Corsi backward block task with higher scores on average for the TD group, and, as expected, on the recalling sentence and word class receptive task, in favor of the TD group.

### 2.2. Tasks and Stimuli

Three tasks taken from the *fLEX* assessment tool ([53]) were used: action naming, sentence production and sentence comprehension. Approximate time to finish the three tasks was 12 min. These are described below.

#### 2.2.1. Action Naming

In the action naming task, stimuli consisted of thirty action verbs: ten intransitive ones (e.g., *tomber* ‘to fall’), ten transitive ones (e.g., *construire* ‘to build’) and ten ditransitive ones (e.g., *donner* ‘to give’) (see Appendix A for the complete list with direct and indirect objects used). The vast majority (80%) of *fLEX* stimuli emerge or are acquired at 30 months of age ([66]) or are listed as part of the first 500 words of French-speaking children in Quebec ([5]). These were frequency-controlled across transitivity types based on 2 online corpora, Lexique ([47]) and Manulex, a corpus from elementary school readers ([41]). No significant differences were found between the three verb groups for syllable length and frequency (see Appendix B).

#### 2.2.2. Sentence Production and Comprehension Tasks

Each task was built using the same sentences involving two manipulations: SV agreement and argument structure complexity. SV agreement was assessed by asking participants to produce or comprehend five intransitive and five transitive verbs—taken from the action naming task—in the third singular or third plural present tense (sentence types 1–4; grey cells in Table 2). These verbs had audible number inflection by means of a coda consonant in the plural and were consonant initially to avoid liaison.

Sensitivity to argument structure complexity was assessed by using five intransitive, five transitive and five ditransitive verbs in singular subject agreement contexts to avoid additional SV agreement processing costs on this feature. In comprehension, argument structure was assessed on these same sentences (sentence types 1, 3 and 5; framed cells in Table 2).

### 2.3. Procedure

Participants were tested individually within one session using the tablet-based fLEX subtasks described above. All sessions were recorded on audio tapes for audit purposes. The examiner (second author or a research assistant) provided oral instructions for the tasks. Before each task, three practice stimuli using extra items were provided with answers. Action naming and sentence production were run first, followed by sentence comprehension, because the same verb stimuli are used in these tasks. This way, one can make sure that the verbs used by the participants in production tasks have not been previously heard during the comprehension task. During the action naming task, participants were instructed to produce an uninflected verb to name an action depicted by a picture on the screen. During sentence production, the visual stimuli were the same, but participants were asked to describe the picture by producing a full sentence in the present tense. In order to elicit this tense, the examiner would prompt participants by asking, for example, “What does this dog do *every day*?” During sentence comprehension, in each trial participants saw four pictures on the screen: one illustrating the target sentence and the three foils—lexical (different action), inflectional (different number of subjects) or mixed (different action and number of subjects)—while listening to a one-sentence description of the target picture through the tablet’s headphones. Participants were instructed to select the picture corresponding to the sentence they heard and to touch it on the screen.

#### Response Coding

Responses were noted directly on paper by the tester during the session. Following this, items were coded according to three criteria: (1) target vs. non-target response, (2) whether the non-target answer was grammatical, and (3) error type produced by the participant (cf. Table 3). A given response could include more than one code if a child made several errors on the same item. A score of 0 was given only for non-target responses that were ungrammatical, semantically inappropriate, or where no output/utterance was provided. Substituting a verb with a different argument structure from the target was counted as grammatical if it semantically matched with the picture (e.g., *chuchoter* ‘whisper’ for *dire* ‘tell’). However, this response type was also counted in our qualitative analysis to check whether their occurrence was more common in the DLD or TD group. In the same way, direct or indirect object omissions were counted as grammatical when the verb was correctly inflected (e.g., *La mère lit à ses enfants* ‘The mother reads to her children’ for the target *La mère lit une histoire à ses enfants* ‘The mother reads a story to her children’). This response type was included in our qualitative analyses.

Coding reliability statistics for participants with DLD were run on the two first coding types only (target response and grammatical answer), since the third one (error type) did not use a nominal scale. The first judge (second or third author) rated production and comprehension during testing. Subsequently, a research assistant (a speech–language pathology (SLP) student or the second author) compiled the participants’ scores into a database. Following this, a second research assistant (SLP student) made a word-for-word transcription of audio recordings on a new scoring sheet. The sentence comprehension task was not double-rated, as it did not involve a verbal response. One participant’s data for the action naming task was not included in the double rating because the audio recording was lost. Marginal homogeneity tables were created for each task to determine scoring trends for both judges ([44]). For the action naming task, agreement between the two judges was perfect. In the sentence production task, coding agreement was high (98.92%). Considering this, we ran the reliability process for target answers on only 20% of the TL group, for which four TL participants were randomly selected. Interrater reliability was 99% for both production tasks.

For the present study, three analyses are presented. First, global results within each task were established. Second, 20 items (Table 2: sentence types 1–4) were used to assess SV agreement in both production and comprehension. Third, 15 stimuli (Table 2: sentence types 1, 3 and 5) were used to assess argument structure effects in action naming and sentence production. Statistical analyses for response accuracy data were run using the *lme4* package for logistic regressions ([4]) in R (version 4.0.3). For target responses, random intercepts for participants and items were included, and the dependent variables were group (two levels), argument structure (three levels), and number (two levels, only in sentence production and comprehension). Age was recoded as a centered continuous variable. We built the models, adding factors such as simple effects and interactions until we found the optimal one, as determined by comparing them using the Anova wrapper (Type III Wald chi-square test) in the *Car* package ([26]). Post hoc pairwise comparisons were performed using the *lsmeans* ([37]), when warranted. Error analyses were run using logistic regressions. Follow-up Mann–Whitney analyses were used to compare error patterns when supported by significant interactions.

## 3. Results

### 3.1. Action Naming

#### 3.1.1. Target Verb Production

For all items in action naming, a first analysis established whether argument structure, participant group or age impacted results. The optimal model for action naming contained group as a simple factor, as well as random effects for participants and items (AIC = 148, BIC = 167.7, log likelihood = −70, nb of observations = 1016). Adding additional factors (including age) or interactions did not improve the model. Participants with DLD had high naming scores (*M* = 97.45, *SD* = 15.17) and performed similarly to the TL group (*M* = 99.41, *SD* = 0.77, *z* = −1.804, *p =* 0.0712). Only one participant with DLD had a low naming score (D03, with 83%).

#### 3.1.2. Non-Target Response Analysis for Verb Production

The optimal model for verb naming errors included argument structure as a principle effect, as well as random effects for items and subjects (AIC = 235.2, BIC = 254.4, log likelihood = −1126, nb. of observations = 340). Adding participant group to the model did not improve it.[note 1] The three frequent non-target response types were (1) verb substitution with the same argument structure (e.g., *mordre* ‘bite’ → *machouiller* ‘chew’), (2) verb substitution with different argument structure (e.g., *dire* ‘tell’ → *parler* ‘speak’) and (3) lexical category changes (e.g., *dire* ‘tell’ → *secret* ‘secret’). Details are provided in Table 4.[note 2]

The effect of transitivity on verb substitutions of the same argument structure was significant (*H* = 65.36, *p* < 0.0001). Intransitive verbs were replaced with verbs of the same argument structure (*n* = 32) as often as transitives (*n* = 58, *p* = 0.062), while ditransitives were more often replaced with verbs with similar argument structures than intransitives (*n* = 167, *U* = 1148, *z* = −6.99, *p* < 0.0001, *d* = 4.31), and transitives (*U* = 118.5, *z* = −6.62, *p* < 0.0001, *d* = 2.82). A different pattern was found on verb substitutions with different argument structures (*H* = 31.36, *p* < 0.0001). Intransitive verbs were less likely be substituted (*n* = 6) than transitives (*n* = 28, *U* = 938, *z* = −4.41, *p* < 0.0001, *d* = 1.53) and ditransitives (*n* = 38, *U* = 990, *z* = −5.05, *p* < 0.0001, *d* = 1.55), which, however, did not differ from each other (*p* = 0.1615). Transitivity effects were not found for lexical category changes, which were rare overall but mainly observed in the DLD group (see Appendix B for details).

### 3.2. Sentence Production

#### 3.2.1. Target Sentence Production

The optimal model for sentence production contained simple effects of group, number, and cAGE, as well as random effects for participants and items (AIC = 979.4, BIC = 1013.1, Table 5). Adding argument structure did not improve the model. Interactions involving principle effects likewise did not result in better models.

We found that participants with DLD had lower sentence production abilities (*M* = 80.71, *SD* = 10.51) than the TL Group (*M* = 96.47, *SD* = 3.97). Sentences with singular targets (*M* = 98.53, *SD* = 12.06) were better produced than those with plural ones (*M* = 74.12, *SD* = 43.86); however, this did not interact significantly with group and appears to be an additive effect, especially for participants with DLD (See Table 6). Although age impacted ability to carry out the task, this effect did not interact significantly with group. Note that this analysis was run on variable verb items only (n = 20) as these items are the ones exhibiting singular–plural distinction in oral French.

#### 3.2.2. Non-Target Response Analysis for Sentence Production

The optimal model for non-target sentence production contained group as a simple effect, as well as random effects for items and participants (AIC = 11.3, BIC = 25.8, log likelihood, −1.6, nb. of observations = 282). Adding other factors or interactions did not improve the model. The five frequent types of responses were (1) verb substitution with the same argument structure (e.g., *mordre* ‘eat’ → *machouiller* ‘chew’), (2) verb substitution with different argument structure (e.g., *dire* ‘tell’ → *parler* ‘speak’), (3) object omission (e.g., *L’homme met dans un verre* ‘The man puts in a glass’), (4) progressive tense (e.g., *Le poulet est en train de rôtir*, ‘the chicken is in the process of roasting’), and (5) verb number agreement errors (e.g., *Les enfants *dort* [dɔʁ] for *dorment* [dɔʁm], ‘The children *sleeps’). Direct or indirect object omission (e.g., *Le garcon lance la balle au chien* ‘the boy throws the ball to the dog’ → *Le garçon lance la balle* ‘the boy throws the ball’), uninflected verbs (e.g., *Les deux messieurs remplir le verre*, ‘The two men to-fill the glass’) and use of other tenses were only produced by participants with DLD but accounted for only few tokens (five or six each; see Table 7 for details).[note 3] Follow-up Mann–Whitney analyses revealed differences between children with DLD and the TL group on verb number agreement errors only (*U* = 269.5, *z* = 4.29, *p* < 0.0001, *d* = 1.957). Grouping argument structure response types (i.e., direct object, indirect object and subject omission) for comparisons did not reveal additional differences between groups.

### 3.3. Sentence Comprehension

#### 3.3.1. Target Sentence Comprehension

The optimal model for sentence comprehension contained a simple effect of group and a non-significant effect trend for number, as well as random effects for participants and items (AIC = 187.9, BIC = 209.6, Table 8).

Again, participants with DLD performed at significantly lower levels than the TL group, but nevertheless showed a high level of comprehension (DLD: *M* = 93.53, *SD* = 6.31; TL group: *M* = 99.41, *SD* = 1.66). Although number effects supported better comprehension of plural (*M* = 98.24, *SD* = 13.19) over singular forms (*M* = 94.71, *SD* = 22.42), this did not significantly interact with participant group. Only participants with DLD made plural comprehension errors. This lack of interaction might be due to low numbers of items or high levels of correct responses. Nonparametric analyses of errors allowed us to probe these error types.

#### 3.3.2. Error Analysis for Sentence Comprehension

Because the number of errors was so low in sentence comprehension, we simply ran nonparametric statistics on errors (in Table 9). A Mann–Whitney analysis revealed that participants with DLD chose inflectional distractors more often than the TL group (*U* = 49, *z* = 3.27, *p* < 0.0005, *d* = 1.205). Lexical or mixed distractors were only observed in the DLD group but were rare. These errors happen more often in the singular (e.g., *il dort* ‘he sleeps, *n* = 14) than the plural (e.g., *ils dorment* ‘they sleep’ *n* = 6), although only participants with DLD make comprehension errors on plural targets.

### 3.4. Additional Analyses on Argument Structure and Subject–Verb Agreement Effects Across Tasks

Because we initially hypothesized that argument structure would impact results, we explored results for the 15 verbs used across both verb naming and sentence production (Table 2: sentence types 1, 3 and 5) and the 20 variable verbs used across sentence comprehension and production (Table 2: sentence types 1–4).

#### 3.4.1. Argument Structure Effects Across Action Naming and Sentence Production Tasks

A logistic regression model with target production for the 15 verbs used in action naming and sentence production was created. The optimal model for target production contained group as a simple factor, as well as random effects for participants and items (AIC = 80.4, BIC = 97.4, log likelihood = −36.2, nb. of observations = 510). No significant differences between groups were found (Estimate: −1.567 SE: 2.185, *z* = −0.717 *p* = 0.47337) and no effect of argument structure was found.

#### 3.4.2. Subject–Verb Agreement Effects Across Sentence Comprehension and Production Tasks

A second logistic regression model for 20 variable verbs used across both sentence comprehension and production tasks was run. The optimal model for target production and comprehension contained group as a simple factor, interaction between number and task, and random effects for participants and items (AIC: 558.1, BIC: 594.6, Table 10). However, these effects did not interact with group.

Post hoc analyses on number effects and sentence production vs. comprehension tasks (Figure 1) reveal that singular items were recognized less accurately in the comprehension task than they were inflected in the production task (*t* 2.64, *p* = 0.0416), while the contrary was found for plural verbs (*t* −7.163, *p* < 0.0001).

## 4. Discussion

In the present study, French teenagers with DLD performed three tasks (1) action naming; (2) sentence production; and (3) sentence comprehension in order to assess whether they exhibit lexical or inflectional verb processing deficits in both production and comprehension. In addition, each task included verbs with different argument structure types (intransitives, transitives and ditransitives) in order to determine whether argument structure complexity increases both lexical and morphosyntactic difficulties. If French teenagers with DLD had performed significantly worse than their peers in sentence production and comprehension tasks, but not in action naming, we assumed this would reflect an underlying *morphosyntactic* deficit but not a lexical one.

The results support this interpretation: in action naming, participants with DLD performed similarly to the TL group and had high scores. We found little evidence of argument structure complexity in our tasks: in the action naming task, i.e., production of uninflected verbs, the three verb types were produced by the two participant groups in equal numbers, and although substitutions for verbs with different argument structures when ditransitive and transitive targets were found, they did not vary according to group. On the other hand, in sentence production, i.e., production of inflected verbs, participants with DLD showed lower abilities than the TL group. In addition, further analyses revealed differences between children with DLD and the TL group on verb number agreement errors only. The predominant error type produced by French teenagers with DLD was subject–verb agreement errors, and this is the only response type on which they significantly differed from the TL group. They were predominant in plural contexts which require an overt verb final consonant in French. Such errors were also produced by TL participants but to a much lesser extent (8% errors in the TL group versus 45% in participants with DLD). Furthermore, although they accounted for only few tokens, uninflected verbs and the use of other tenses were response types produced only by participants with DLD (mirroring younger children’s patterns, e.g., [61]). Likewise, in sentence comprehension, participants with DLD performed at significantly lower levels than TL participants, but nevertheless showed good comprehension abilities, which were noticeably better than their production abilities. Interestingly, while further analyses of sentence production revealed better production of singular over plural verb forms, the reverse pattern was observed in comprehension, where more errors were found for singular targets. Almost all comprehension errors were morphosyntactic (i.e., pointing to the inflectional distractor rather than to the lexical or the mixed one). Comprehension errors did not increase with argument structure complexity nor did the number of sentence production errors or SV agreement errors. It could be that age effects are influencing results, as the sentence production task showed a simple effect of age on results. However, this appears not to be related to the fact that the TL control group was on average younger than the DLD group, as even the youngest TL participants showed very low levels of error. In fact, the effect of age is such that as age goes up scores fall. This appears to be a corollary of the DLD group being older on average than the TL group. If the TL group had been matched on age with their peers with DLD, we surmise that group differences would have been even stronger than those we found.

Since argument structure complexity effects have been reported in French children with DLD ([29]; [51]), we expected more errors to be produced in producing ditransitive verbs than transitive or intransitive verbs. However, verb argument structure complexity had no impact on results from our action naming and sentence production tasks. This might be because our participants were slightly older than those who participated in the cited study (our age range was 12–15 while Pizzioli & Schelstraete’s age range was 8–13 (*M* = 9;11; *SD* =1.7). More importantly, they used only four transitive and four intransitive verbs to evaluate transitivity effects. Our data, based on a larger set of verbs, reveal that teenagers with DLD have relatively strong syntactic abilities and are able to build structures involving zero, one, or two post-verbal complements. Our data suggests that while SV agreement errors persist in teenagers with DLD, argument structure deficits do not. However, in order to confirm this, a larger group of participants should be assessed in a follow-up study using similar tasks.

Our study establishes that SV agreement errors are a linguistic feature of French teenagers with DLD. These results support the claim that verb inflection errors persist in teenagers with DLD, as reported for Dutch speakers by [21] ([21], [22]). The underlying deficit appears to be morphosyntactic rather than lexical, since a similar performance was observed in action naming for both groups but a different performance was observed in sentence production. More precisely, verb inflection seems particularly impaired since plural SV agreement production—much more saliently than comprehension—is still vulnerable for teenagers with DLD.

In comprehension, although counterintuitive, singular verb forms were more problematic than plural verb forms. Other studies on SV agreement comprehension have shown an advantage for plural forms in comparison to singular forms in Spanish ([50]). This was attributed to the phonological salience of morphemes, which affects comprehension but has no impact on production and may lead to opposite patterns across languages (for instance in Spanish vs. English). [54] ([54]) also revealed that, in Spanish, sentence comprehension scores were better in plural than in singular SV agreement in both Basque–Spanish balanced and Spanish-dominant bilingual children (age range: 5–10; *M* = 7;1 and 7;5, respectively). According to these authors, this reflects the impact of language-specific morphological features on sentence production and comprehension: whereas an overt verb inflection may increase morphosyntactic production difficulties, it may facilitate morphosyntactic comprehension by virtue of being more salient and thus more easily perceivable. In Italian, [40] ([40]) also reported that a group of children with DLD aged 3;8–5;7 made more errors than the TL group understanding plural *and singular* verbs in the third person, in contrast to production data where singulars appeared to be mastered. As assumed by the Morphological Richness Account ([11]), the acoustic properties of agreement markers may influence their mastery. Unlike in English, in French, third singular subject–verb agreement is phonologically less salient than third plural subject–verb agreement (e.g., *Il mord* [ilmɔ**ʁ**] *l’os* ‘He bites the bone’ vs. Ils *mordent* [ilmo**ʁd**] *l’os* ‘They bite the bone’). Therefore, this salience in itself may explain why, in the current study on French DLD manifestations in teenagers, singulars were better produced than plurals in sentence production (*t* 6.985, *p* < 0.0001) but were not better recognized than plurals in sentence comprehension (*t* −2.401, *p* = 0.0774) (Cf. Figure 1).

ERP data from our groups also support group differences in singular SV agreement error processing using auditory–visual mismatches, where a grammatical singular verb stimulus was paired with an image of a plural subject (e.g., “*In the evening*, *the boy sleeps in the bed’* with an image of two boys sleeping). For these errors, the TD group elicited an ERP component (a late positivity, indicating error recognition and repair), while the group with DLD elicited no ERP signature, signaling the absence of a cognitive response to singular agreement errors for plural visual stimuli. In contrast, plural errors for singular visual stimuli elicited a similar N400s (a lexico-semantic ERP component) in both groups ([16]).

The present study thus adds data from French and from teenagers supporting the view that it might be more problematic to decode singular rather than plural inflected verb forms. This also supports the relevance of assessing both production and comprehension, as they might lead to diverging patterns. We must note, however, that only relatively high-frequency verbs were used in our tasks. It could be that comprehension of lower-frequency plural forms might also result in difficulties in teenagers with DLD.

The Procedural Deficit Hypothesis ([68]), which assumes relative lexico-semantic strengths but impaired grammatical computation, specifically on aspects learned through procedural memory, can explain the data reported in the present study, if we consider that inflected verb forms are not lexicalized but built online, as suggested by the Dual Route Model of morphology. This would explain the significantly lower scores observed in sentence production than in action naming. This would also explain the higher performance in comprehension than in production, and why argument structure complexity, which is assumed to rely on declarative memory, appears to have no impact on verb inflection encoding difficulties. However, according to this hypothesis, if irregular verb forms are lexicalized, they should not be problematic. Therefore, our results partially confirm the Procedural Deficit Hypothesis since errors were made on irregular verbs too (e.g., *les jeunes garçons *s’endort* ‘the young boys *falls asleep’). Note that [62] ([62]) found that first-graders with DLD show no effect of verb conjugation group, i.e., low scores on all types of past participle forms. The Extended Optional Infinitive account ([57]) is likewise not supported by our data, as only two occurrences of infinitive verbs were found in sentence production. Our data rather support the Extended Optional Default Account ([48]), the term “default” instead of “infinitive” meaning there might be an overuse of the singular present finite “default” verb form in French children with DLD. It is important to note, however, that this account explains morphosyntactic deficits as observed for verb agreement. Teenagers with DLD, including our participants, show deficits in other linguistic domains such as more complex tasks involving lexical–semantic relations and sentence repetition involving multiple domains of language including aspects of morphosyntax not addressed here ([16]), going beyond predictions made by this account. Finally, our results support the Fragile Computation of Agreement Hypothesis ([46]), which assumes a specific impairment in abstract grammatical computation, which underlies both production and comprehension.

The generalizability of our study is limited by our small number of participants. Furthermore, our 30 verbs are frequent and early-acquired, which does not allow for an exhaustive assessment of verb production and comprehension, and it could be that relative strengths (verb naming, thematic structure) would have disappeared if less frequent verbs had been used, as has been found for TD children in oral language and writing tasks (e.g., [60]; [33]). To further our knowledge of verb production and comprehension deficits in DLD, future studies could integrate verb frequency as a factor.

From a clinical perspective, the present study presents new data from French teenagers with DLD showing that verb inflection production errors persist in adolescence in production and comprehension, and that depending on language-specific morphological properties, they may affect singular or plural subject–verb agreement in different ways. The comprehension task is also a quick and efficient way to assess subtle inflectional impairments. As the action naming performance is similar in DLD and in the TL group, the current action naming task of the fLEX test is not sufficient to identify verb production difficulties in DLD, unless it includes a more exhaustive list of verbs with different degrees of frequency. Also, other properties of verbs such as regularity should be considered for oral language assessment, in addition to the usual properties such as phonological complexity and lexical frequency, because they might be an additional source of difficulty. Nevertheless, we believe that the study of teenagers has the advantage of highlighting a partial picture of strengths and weaknesses in French-speaking teenagers with DLD, most notably in the domain of morphosyntax, given their greater number of years of experience with language than preschoolers.

## 5. Conclusions

Subject–verb agreement difficulties appear to persist in French-speaking teenagers both in sentence production and sentence comprehension, despite accurate lexico-semantic knowledge of these same verbs. Although typically developing speakers of French are also still consolidating their grammars, as seen by errors in sentence production, the sentence production task clearly distinguishes children with DLD from the TL participants. Subtle comprehension deficits are also observed in teenagers with DLD. These results are consistent with the Fragile Computation of Agreement hypothesis, which predicts agreement deficits in both production and comprehension. They represent new data from DLD manifestations in French-speaking teenagers.

## 6. Patents

The intellectual property of the fLEX tool ([53]) funded by Marie Curie action FP7-PEOPLE-2011-IOF was registered at the territorial registry of intellectual property of the Basque Country (01/2016/88) and the University of Montreal (DGR 03060, 11 October 2014).

## Figures and Tables

**Figure 1 behavsci-15-01252-f001:**
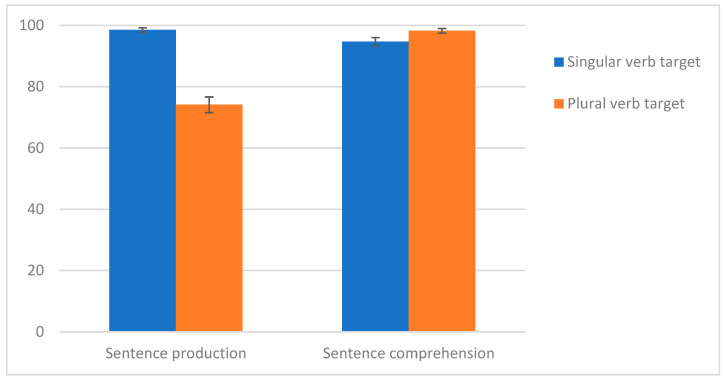
Verb number effects on sentence comprehension and production. Sentence production was significantly lower than comprehension. Significant differences between singular and plural verbs were found, in favor of plural in comprehension versus production, and in favor of the singular in production versus comprehension.

**Table 1 behavsci-15-01252-t001:** Participant characteristics. Comparisons between groups are expressed as the Brunner–Munzel statistic (t_bm_), a *p*-value and a Common Language Effect Size (CLES), indicating the probability of a random observation from the DLD group being larger than a random observation from the TL group, with 0.5 being at chance.

	DLD Group (N = 17)		TL Group (N = 17)		Brunner–Munzel Tests		
	Mean	SD	Mean	SD	t_bm_	*p*-Value	CLES
Age	14.06	0.68	12.76	1.49	13.77	<0.001	0.79
School	7.53	0.51	6.59	1.50	7.43	<0.001	0.72
Sent Rec	54.69	7.82	69.69	8.88	43.02	<0.0001	0.90
Word Rec	24.0	22.37	66.71	22.60	9.20	<0.0001	0.08
Corsi–F	5.56	1.55	5.94	1.44	0.60	0.55	0.44
Corsi–B	4.94	1.06	5.94	1.68	2.10	**0.04**	0.31
DMTS–1s	0.88	0.10	0.92	0.07	1.24	0.22	0.37
DMTS–5s	0.84	0.13	0.87	0.09	0.57	0.57	0.44

Notes: Chronological age (Age) and schooling (School) are expressed in years. Recalling sentences (Sent Rec) and word class receptive (Word Rec) CELF-IV^cnd-F^ scores are untransformed; Corsi block scores reflect forward (Corsi–F) and backward (Corsi–B) spatial spans; and delayed match-to-sample represents the accuracy for 1 s (DMTS–1s) and 5 s (DMTS–5s) delays.

**Table 2 behavsci-15-01252-t002:** Sentence types used in the production and comprehension tasks (indexed with checkmarks), classified by transitivity and agreement type. The crucial morphological elements are in bold. Grey shading indicates singular–plural contrasts, while framed cells indicate transitivity contrasts.

		SV Agreement	Transitivity
Type	Example	Sg	Pl	In	Tr	Di
1	Le monsieur/Il **part** [ilpaʁ]‘The man/He leaves’	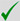		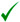		
2	Les messieurs/Ils **partent** [ilpaʁt]‘The men/They leave’		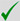	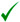		
3	Le chien/Il **mord** [ilmɔʁ] l’os‘The dog/He bites the bone’	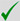			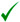	
4	Les chiens/Ils **mordent** [ilmoʁd] l’os‘The dogs/They bite the bone’		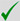		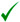	
5	L’enfant **lance** la balle au chien.The child throws the ball to the dog	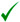				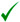

**Table 3 behavsci-15-01252-t003:** Type of non-target answer in fLEX, according to task (1 = accepted; 0 = unaccepted response).

Task	Type of Error	Code
2 & 3	Substitution with another lexical category (e.g., *secret* ‘secret’ for *parler/dire* ‘speak/say’)	0
2 & 3	Unintelligible or inexistent word	0
2 & 3	Substitution with a verb of same transitivity (e.g., *lire* ‘read to’ for *parler/dire* ‘speak/say to’)	1
2 & 3	Substitution with a verb of different transitivity (e.g., *chuchoter ‘whisper’* for *parler/dire* ‘speak/say’)	1
3	Substitution with an uninflected verb (e.g., *dormir* ‘to sleep’ for *ils dorment* ‘they sleep’)	0
3	Subject omission (e.g., *_ construit la maison* ‘_ builds the house’)	0
3	Over-regularisation (e.g., *senté* [sãte] for *senti* [sãt^s^i] ’smelled’)	0
3	Direct object omission (e.g., *les deux gars sentent_ *‘the two boys smell _’)	1
3	Use of a different tense (e.g., perfect past: *s’est endormi* ‘fell asleep’).	1
3	Substitution with a progressive form (e.g., *en train de dire* ‘in the process of saying’ = ‘is saying’)	1
3 & 4	Verb agreement error (e.g., *grandit* [grãd^z^i] ‘grows’ for *grandissent* [grãd^z^is] ‘grow’)	0
3 & 4	Lexical error (e.g., image of *nage* ‘(he) swims’ for *grandit* ‘(he) grows’)	0
3 & 4	Mixed error (e.g., image of *nagent* ‘(they) swim’ for *grandit* ‘(he) grows’)	0

Notes: Task 2 = action naming; Task 3 = sentence production; Task 4 = sentence comprehension.

**Table 4 behavsci-15-01252-t004:** Total number of response types for action naming by verb transitivity (in parentheses: number of children producing them), means, medians and ranges. Results are not divided by participant group, since the model did not support group effects on this task. However, Appendix B provides this information.

Response Types	Total	Mean	Median	Range
*Instransitive verbs*				
Verb substitution (same ArgStr)	32 (24)	0.94	1	0–3
Verb substitution (diff. ArgStr)	6 (6)	0.18	0	0–1
Lexical category change	7 (7)	0.21	0	0–1
*Transitive verbs*				
Verb substitution (same AgStr)	58 (29)	1.71	1.5	0–4
Verb substitution (diff. ArgStr)	28 (27)	0.82	1	0–2
Lexical category change	2 (2)	0.06	0	0–1
*Ditransitive verbs*				
Verb substitution (same AgStr)	167 (34)	4.91	5	2–6
Verb substitution (diff. ArgStr)	38 (29)	1.12	1	0–4
Lexical category change	5 (4)	0.15	0	0–2

**Table 5 behavsci-15-01252-t005:** Model of sentence production effects for participant group, cAGE, and number (random effect structure: participants and items).

Estimate	Std.	Error	z	Pr (>|z|)
Intercept: TL froup, singular verbs	5.6358	0.4918	11.460	<0.000001
Group: DLD	−2.0661	0.4217	−4.900	<0.000001
cAGE	−0.6165	0.2699	−2.284	=0.0224
Plural verbs	−1.5586	0.3041	−5.126	<0.000001

Log likelihood = −483.7.2; Nb. of observations = 2040.

**Table 6 behavsci-15-01252-t006:** Response accuracy (means and standard deviations) by number and group for the sentence production task.

	Singular	Plural
TL Group	100 (0)	92.36 (0.27)
DLD Group	97.06 (0.17)	55.88 (0.50)

**Table 7 behavsci-15-01252-t007:** Total number of response types (number of children producing them), means, medians and ranges for the sentence production task. Bold type indicates significant group differences.

Response Types	Total	Mean	Median	Range
*DLD Group*				
Verb substitution (same ArgStr)	40 (16)	2.35	1	0–4
Verb substitution (different ArgStr)	30 (14)	1.76	2	0–4
Progressive tense	11 (8)	0.65	0	0–2
**Verb number agreement**	**75 (16)**	**4.41**	**4**	**0–9**
Direct object omission	5 (4)	0.29	0	0–2
Indirect object omission	6 (6)	0.35	0	0–1
Uninflected verb ^a^	5 (1)			
Other tense ^a^	5 (4)	0.29	0	0–2
*TL Group*				
Verb substitution (same ArgStr)	47 (17)	2.76	3	1–6
Verb substitution (different AgStr)	24 (13)	1.41	1	0–4
Progressive tense	13 (9)	0.76	1	0–2
**Verb number agreement**	**13 (8)**	**0.76**	**0**	**0–3**
Direct object omission	2 (1)			
Indirect object omission	2 (1)			

^a^ Only produced by participants with DLD.

**Table 8 behavsci-15-01252-t008:** Model of effects for participant group and number (random effect structure: participants and items) on sentence comprehension.

Estimate	Std.	Error	z	Pr (>|z|)
Intercept: TL group, singular	5.1335	0.8306	6.180	<0.000001
Group: DLD	−2.5420	0.7765	−3.274	=0.00106
Number: plural	1.233	0.6336	1.946	=0.05161

Log likelihood = −88.5; Nb. of observations = 680.

**Table 9 behavsci-15-01252-t009:** Total number of response types (number of children producing them), means, medians and ranges for the sentence comprehension task.

Response types ^a^ Only Evidenced in Participants with DLD	Total	Mean	Median	Range
*DLD group*				
**Inflectional distractor**	**19 (12)**	**1.12**	**1**	**0–5**
Lexical distractor ^a^	2 (2)	0.12	0	0–1
Mixed distractor ^a^	1 (1)			
*TD group*				
**Inflectional Distractor**	**1 (1)**	**0.06**	**0**	**0–1**

^a^ Only produced by participants with DLD.

**Table 10 behavsci-15-01252-t010:** Model of effects for participant group, number and task (random effect structure: participants and items) on sentence comprehension.

Estimate	Std.	Error	z	Pr (>|z|)
Intercept: TL group, singular, Task 3 sentence production	9.8010	1.0548	9.292	<0.000001
Task4: comprehension	−6.2069	1.2079	−5.139	<0.000001
Group: DLD	−2.4602	0.3779	−6.510	<0.000001
Number: Task 4	4.7827	0.7406	6.458	<0.000001

Log likelihood = −272.1; Nb. of observations = 1360.

## Data Availability

Data are available upon request.

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
