# Peer review of "Investigation of Lexical and Inflectional Verb Production and Comprehension in French-Speaking Teenagers with Developmental Language Disorders (DLDs)"

_behavsci, 2025, doi:10.3390/bs15091252_

Round 1
Reviewer 1 Report
Comments and Suggestions for Authors
The study is a valuable contribution to the field of DLD since it investigates adolescents using three measures targeting lexical semantics and morphosyntax. In addition, it teases apart comprehension and production. The reviewed background literature is quite rich, the statistical analyses are sufficiently sound, the discussion mentions limitations. However, there are major issues with the terminology used and the way the content is presented, discussed as well as argued (at times missing discussion and contradictory argumentation) that give rise to ambiguity and confusion on the part of the reader. Accordingly, for publication, much work has to be done in terms of serious reordering and streamlining. In the following, I provide some examples:
1) As to the terms used for the linguistic features investigated, you simply use too many for the same: for problems with the lexicon these are e.g. difficulties with lexical retrieval, lexical access, lexical semantics, lexical deficits, for problems with language structure these are difficulties with morphosyntax, with inflection, with producing inflected verbs, with inflectional processes, with verb morphological encoding, inflectional processing deficits, verb inflection deficits, verb processing deficits etc.
2) Relatedly, at times the terms "representation" and "processing" are addressed but the two are not sufficiently differentiated throughout the paper.
3) In addition, "adolescents", "older children", "teenagers" and "adults" are often referred to without sufficient differentiation
4) As to the presentation of content, the title of section 1.1 contains "across languages" but so do sections 1.2, 1.3 and 1.4.
5) Similarly, section 1.3 is titled "Language production and comprehension" but only discusses "comprehension"
6) Furthermore, section 1.3 contains accounts on DLD that deserve a subsection on their own. In addition, accounts on DLD are mentioned in section 1.4 and elsewhere. Overall, this content is quite scattered, not adequately discussed in a comparative manner and much of it is not taken up at all thereafter.
7) The content on previous studies as well as accounts on DLD are not adequately taken up in the considerations of the present study and the same goes for the discussion of the article.
8) Some of the interpretation of results is unclearly described - e.g. "perform similarly to controls despite numerical differences" (lines 445-446).
9) Relatedly, it remains unclear whether the difference between the production and the comprehension task in terms of the rates of the participants with DLD is a statistical one or a "numeric" one - this is crucial for one of the main claims of the article which is that comprehension is better than production ("performed at lower levers than those without, but nevertheless showed good comprehension abilities, which was noticeably better than their production abilities, lines 592-593).
10) In addition, in the abstract and in the section, in which the results are presented you claim that participants with DLD have problems with both comprehension and production but this is contradictory to one of the claims in the discussion, namely that "SV agreement production - but not comprehension- is still vulnerable for teenagers with DLD", lines 619-621). Accordingly, if some of the differences mentioned are not significant, this has to be clearly stated so that misinterpretation can be successfully avoided.
Comments on the Quality of English LanguageHere, just one example for an issue of expression:
"However other languages have been investigated including non-Indo-European and Romance languages such as Italian and French." (lines 48-49) - this might be read as if Italian and French are not Indo-European.
Reviewer 2 Report
Comments and Suggestions for Authors
This article included 17 DLD and 17 controls to investigated the verb inflection and argument structure processing difficulties in French-speaking adolescents. They found that French-speaking adolescents with developmental language disorder (DLD) continue to show significant difficulties with verb inflection, particularly subject–verb agreement, over-relying on singular default forms in production and displaying heightened sensitivity to singular verb forms in comprehension. Notably, they performed comparably to typically developing peers on action naming—indicating intact lexical access—while argument-structure complexity did not affect task performance, suggesting that the core deficit in adolescent DLD lies in morphosyntactic processing rather than in lexical retrieval or argument-structure representation. The topic is interesting and to check if the DLD persist over time. The data collection and anlyses sounds reasonable, and the results is reliable. I would recommend accepting after several concerns addressed:
- The literature review is overly fragmented and lacks focus. It is recommended to make it more concise.
- Given the small sample size, the authors should use temper claims about “no effect of argument-structure complexity”, and it could be framed as “we found no evidence for an effect within the limits of our design” rather than evidence of absence.
- The DLD group is on average 1.5 years older and one school-year ahead of controls (Table 1). These differences are statistically significant and could partially account for the observed similarities in action naming. The manuscript should either (a) statistically control for age and schooling in all analyses, or (b) discuss how residual age-related advantages might have masked subtle deficits.
- These differences are statistically significant and could partially account for the observed similarities in action naming. The manuscript should either (a) statistically control for age and schooling in all analyses, or (b) discuss how residual age-related advantages might have masked subtle deficits.
- Several post-hoc comparisons (e.g., the singular/plural interaction in Figure 1) lack correction for multiple testing. Adjusted p-values or Bayes factors should be provided.
Reviewer 3 Report
Comments and Suggestions for Authors
Overall summary
This study examined how French-speaking teens with developmental language disorders (DLD) understand and utilize verbs, focusing on how well they handle verb inflection and sentence structure. While participants named actions just as well as their peers, they made more mistakes when using verbs in sentences, especially with SV agreement. This adds to existing literature on language pedagogy and how function vs form can interact. Overall, this study is quite well written and adds to literature on language processing and acquisition in a way that is novel.
Overall comments
One strength of this study is its well-structured and linguistically grounded design, which integrates action naming, sentence production, and comprehension to isolate grammatical deficits from lexical ones. Another strength is the use of age-appropriate tasks and stimuli (via the fLEX tool) along with robust statistical analysis and careful error coding, which strengthens the finding that subject-verb (SV) agreement remains a key area of difficulty for adolescents with DLD. However, a major area for improvement lies in the limited sample size and the restricted verb set, which may have masked other important linguistic challenges, especially in lower-frequency or less salient verb forms. I also think there could have been a slightly stronger discussion on how this generalizes to language pedagogy and perhaps even bilingualism more, but I think it’s okay given the scope.
Major Comments / Issues
- On page 19, lines 676–682, the authors themselves acknowledge the small sample size and limited range of verbs. This limits how confidently we can apply these findings to a broader population of French-speaking adolescents with DLD. Could you touch on this a bit more and how one might predict frequency effects?
- As noted on page 18, lines 677–678, the study only included high-frequency verbs. This may have overestimated participants’ abilities in both production and comprehension, and may not reflect difficulties with more complex or rare verbs. I think it’s important to look at frequency as an effect here (similar to my comment above).
- On page 7, lines 331–334, it is revealed that the DLD group was older than the control group, which may introduce confounding developmental effects that aren’t fully accounted for. Could you please address this? Is this a concern for DLD?
- The authors conclude (page 18, lines 611–613) that argument structure complexity did not impact performance. However, this claim may be premature given the relatively small number of items testing this and the broad age range of participants. Please expand a bit or perhaps reduce the strength of this claim.
- The discussion references ERP findings (page 18, lines 640–644) from another study to support processing differences, but these seem incongruent with the text. Perhaps remove or add some slight details. I’m not sure it’s needed here.
Minor comments/typos
- Page 2, Line 83: “and appears to be linked to task and age” << What appears? DLD? Please clarify
- Page 6, Lines 287-288: “Their mother tongue was French”. Please state dominant/L1 (preferred over ‘mother tongue’).
- Page 17, Figure 1 caption: The figure lacks a full sentence caption or description. It currently just says “Interaction of verb number effects on sentence comprehension and production” << What is the statistically significant finding? The error bars are confusing as they overlap.
